# The High Frequency of a G-Allele Variant of the *FOXP3* Gene in Old Asian Cattle Breeds, Water Buffaloes, and Holstein Friesian Cows: A Potential Link to Infertility

**DOI:** 10.3390/ani15162407

**Published:** 2025-08-16

**Authors:** Abdullah Al Faruq, Oky Setyo Widodo, Mitsuhiro Takagi, Tita Damayanti Lestari, Muhammad Fadhlullah Mursalim, Nanang Tedjo Laksono, Hiroaki Okawa, Md Shafiqul Islam, Shinichiro Maki, Tofazzal Md Rakib, Akira Yabuki, Osamu Yamato

**Affiliations:** 1Laboratory of Clinical Pathology, Joint Faculty of Veterinary Medicine, Kagoshima University, Kagoshima 890-0065, Japan; faruqabdullahal103@gmail.com (A.A.F.); si.mamun@ymail.com (M.S.I.); k6993382@kadai.jp (S.M.); rakibtofazzal367@gmail.com (T.M.R.); yabu@vet.kagoshima-u.ac.jp (A.Y.); 2Faculty of Veterinary Medicine, Chattogram Veterinary and Animal Sciences University, Khulshi, Chattogram 4225, Bangladesh; 3Laboratory of Theriogenology, Joint Faculty of Veterinary Medicine, Yamaguchi University, Yamaguchi 753-8515, Japan; oky.widodo@fkh.unair.ac.id (O.S.W.); mtakagi@yamaguchi-u.ac.jp (M.T.); 4Faculty of Veterinary Medicine, Universitas Airlangga, Surabaya 60155, Indonesia; titadlestari@fkh.unair.ac.id; 5Study Program of Veterinary Medicine, Faculty of Medicine, Hasanuddin University, Makassar 90245, Indonesia; dullahmursalim@gmail.com; 6Prigen Conservation Breeding Ark-Taman Safari, Pasuruan 67157, Indonesia; nanangvet_ts2@tamansafari.com; 7Guardian Co., Ltd., Kagoshima 890-0033, Japan; okawa0117@guardian-vet.com

**Keywords:** bovine *FOXP3* gene, regulatory T (Treg) cell, Bali cattle, Jaliteng, water buffalo, Holstein Friesian cow, anti-Müllerian hormone, reproductive failure, infertility

## Abstract

This study investigated the genotypes of an X-linked single-nucleotide variant in the bovine *FOXP3* gene in herds of Bali cattle, Jaliteng cattle, and water buffaloes. This study also investigated the relationship between genotypes and serum concentrations of anti-Müllerian hormone in parous and non-parous Holstein Friesian cows. The G allele frequency was markedly high in Bali (0.944) and Jaliteng cattle (0.714) and water buffaloes (1) and moderately high in Holstein Friesian cattle (0.514). Anti-Müllerian hormone levels were significantly lower (*p* < 0.05) in parous Holstein Friesian cows carrying the G allele than in parous cows carrying the A/A genotype. Our results suggest that the G allele of the *FOXP3* gene variant may be originally a wild-type variant in cattle breeds and water buffalo and that the G allele may be associated with infertility in cattle and other bovid species.

## 1. Introduction

Reproductive failure in cattle is a significant challenge in dairy and beef cattle industries worldwide [1,2]. This stems from a complex combination of managerial, environmental, physiological, and genetic factors. These factors influence various aspects, including selective breeding, postpartum conditions, calving intervals, and the occurrence of healthy calf births [3]. Maternal factors, including age, oocyte defects, endocrinological dysfunction, nutritional abnormalities, genital tract infections, and genetic alterations, are recognized as important factors contributing to infertility [4]. Repeat breeding is a major issue affecting fertility as it leads to reduced fertility in lactating dairy cows on a global scale [5]. Understanding the genetic foundation of reproductive failure is crucial for enhancing herd fertility, rather than individual fertility [4]. Recently, both human and animal studies have increasingly focused on exploring the roles of maternal immunoregulatory cells and their controlling genes to determine their potential involvement in infertility [6,7].

The *FOXP3* gene encodes a forkhead box P3 protein that plays a critical role in regulating the development and function of regulatory T cells as a master regulator, which is important for regulating the immune system and suppressing excessive immune responses [6,7]. Dysregulation of the immune system or an excessive immune response can lead to embryo rejection or disruption of normal reproductive processes, which in turn contribute to infertility [8]. In humans, mutations in the *FOXP3* gene are associated with poor embryonic development, abnormal ovarian follicle development, and poor oocyte quality, all of which contribute to reduced fertility in females [9]; furthermore, in males, mutations in the *FOXP3* gene are associated with testicular inflammation [10].

In cattle, the *FOXP3* gene plays an important role in regulating immune responses and immune system development and is believed to be associated with infertility in bovine species [11,12]. Recently, a genome-wide association study (GWAS) found an association between recurrent infertility, such as repeat breeders, and an X-linked maternal single-nucleotide variant located 2175 base pairs upstream of the start codon of the bovine *FOXP3* gene (NC_037357.1: g.87298881A>G, rs135720414) in Japanese Black (JB; *Bos taurus*) cows with lower *FOXP3* transcript levels [11].

Accordingly, this variant may be a useful target for initiatives to increase cattle herd fertility because of the relationship between a higher frequency of the G allele variant of the *FOXP3* gene and recurrent infertility in JB cows [11]. Our research team preliminarily investigated frequencies of the G allele in various cattle breeds including JB, Holstein Friesian (HF; *B. taurus*), Korean Hanwoo (KH; *B. taurus coreanae*), and Indonesian Madura (IM; a crossbreed between *B. indicus* and *B. javanicus*) cattle and found a higher frequency of G allele in the local Asian cattle breed IM (0.700), a moderate frequency in the dairy cattle HF (0.466), and low frequencies in the beef cattle breeds JB (0.250) and KH (0.112) [12].

Thus, it would be crucial to evaluate the associations between high frequencies of the G allele and infertility in different breeds and bovid species. Old Asian cattle breeds and water buffaloes (*Bubalus bubalis*) in Indonesia have long suffered from subclinical infertility and repeat-breeding problems. These old Asian cattle breeds include Bali cattle (*B. javanicus domesticus*) and Jaliteng cattle, a crossbreed between Bali and banteng cattle (*B. javanicus*), which is a wild bovine species found in Southeast Asia. We hypothesized that the frequency of the *FOXP3* gene variant may be associated with infertility in these cattle breeds and species because one of the Asian local cattle, IM, has a high frequency (0.700) of the G allele variant [12]. Therefore, we performed genotyping of the *FOXP3* gene variants in herds of these old Asian breeds and species.

Furthermore, we investigated how the *FOXP3* gene variant is associated with the peripheral concentration of anti-Müllerian hormone (AMH), a biomarker of fertility that indicates the size of the ovarian reserve [13,14]. To this end, we also performed genotyping of the variant in a herd of HF cows in which AMH concentration was measured and investigated the relationship between the genotypes and AMH concentration in parous and non-parous HF cows.

## 2. Materials and Methods

### 2.1. Animals and Blood Collection

In 2023, blood samples were collected via jugular or caudal venipuncture from 48 Bali cattle (7 males, 41 females; 1–12 years old and over, mean 6.7 years), 5 Jaliteng cattle (3 males, 2 females; 2–10 years old, mean age 6.6 years), and 20 water buffaloes (5 males, 15 females; 3–12 years old, mean 7.2 years). Blood was collected to genotype *FOXP3* variants. The parity of females was 1–16 (mean 4.3) times in Bali cattle, 6–8 (mean 7.0) times in Jaliteng cattle, and 1–5 (mean 3.3) times in water buffalo. Bali cattle were housed in South Sulawesi and East Java Provinces, Indonesia. Jaliteng cattle were housed in East Java Province, Indonesia. Water buffaloes were housed in South Sulawesi Province, Indonesia. Representative images of the Bali and Jaliteng cattle, and water buffaloes, are shown in Figure 1.

In 2019, blood samples were collected via jugular or caudal venipuncture from 69 parous and 39 non-parous HF cows housed in Fukuoka Prefecture, Japan. Blood was collected to measure AMH concentration and to genotype the *FOXP3* variant in parous cows 4 weeks after delivery and in non-parous cows during pregnancy. The parity of the parous cows ranged from 1 to 7 (average 3.2) times. For the measurement of AMH concentration, serum was obtained by centrifugation immediately after blood collection and was stored at −30 °C until measurement.

### 2.2. DNA Extraction and Genotyping

The collected blood was spotted onto Flinders Technology Associates filter paper (FTA card) (QIAcard FTA Classic; Qiagen, Hilden, Germany) and stored at room temperature or in a refrigerator (4 °C) until DNA extraction. DNA was extracted from discs punched out of the blood-spotted FTA cards following appropriate treatment as previously described [15].

Genotyping of the bovine *FOXP3* gene variant (NC_037357.1: g.87298881A>G, rs135720414) was performed as previously described [12] using the extracted DNA as a template. PCR amplifications were performed using a StepOne Real-Time PCR System (Applied Biosystems, Foster City, CA, USA). Specific primer pairs (forward, CCATGTGGCTTCTGAGAAATAGTCA and reverse, TACCTGGAGGGCCAGACT) and TaqMan minor groove binder probes (A-allele, TCTTCCTGCATTGTCTG and G-allele, TCTTCCTGCACTGTCTG), linked to each fluorescent reporter dye (6-carboxyrhodamine and 6-carboxyfluorescein, respectively) at the 5′ end and a non-fluorescent quencher dye at the 3′ end, were used. Real-time PCR amplifications were carried out in a final volume of 10 µL consisting of 2× PCR master mix (TaqMan GTXpress Master Mix; Applied Biosystems) and 80× genotyping assay mix (TaqMan SNP Genotyping Assays; Applied Biosystems) containing the specific primers at 450 nM, TaqMan probes at 100 nM, and template DNA. A negative control containing nuclease-free water instead of the template DNA was included in each run. The cycling conditions were 20 s at 95 °C, followed by 50 cycles of 3 s at 95 °C and 20 s at 60 °C, with a subsequent holding stage at 25 °C for 30 s. The data obtained were analyzed using StepOne version 2.3 (Applied Biosystems).

Female animals were categorized into three genotypes (A/A, A/G, and G/G), and male animals were categorized into two types of hemizygotes (A/- and G/-). G allele frequency was calculated based on the number of G alleles among the total number of X chromosomes in each herd.

DNA samples with three genotypes (A/A, A/G, and G/G or G/-) from 1 HF (1 female), 5 Bali (2 males, 3 females), and 3 Jaliteng cattle (1 male, 2 females) and 2 water buffaloes (1 male, 1 female) were used to validate the genotyping assay, following genotype confirmation based on Sanger sequencing (Kazusa Genome Technologies Ltd., Kisarazu, Japan). For Sanger sequencing, PCR was performed using a forward primer (AGGGCTCAGATGCAGAC) and a reverse primer (GGATATGGTCTGTCTGGT), which produced a 166 base pair amplicon [12]. Genotyping using the real-time PCR assay was performed in 48 Bali cattle (7 males, 41 females), 5 Jaliteng cattle (3 males, 2 females), 20 water buffaloes (5 males, 15 females), and 108 HF cows.

### 2.3. Measurement of AMH

AMH concentration was measured using a bovine AMH ELISA kit (AnshLabs, Webster, TX, USA) according to the manufacturer’s instructions [16]. Undiluted serum (50 µL) was used in the assay. The assay’s limit of detection was 11 pg/mL, and the coefficient of variation was 2.9% according to the manufacturer.

### 2.4. Statistical Analysis

Statistical analyses were performed using R version 4.5.0. Differences in AMH concentrations among the three genotypes were analyzed using the Friedman rank sum test and Wilcoxon rank sum exact test with Bonferroni correction. Differences at *p* < 0.05 were considered statistically significant.

## 3. Results

### 3.1. Allele Frequencies of the FOXP3 Gene Variant

We surveyed 48 Bali and 5 Jaliteng cattle and 20 water buffaloes in Indonesia and 108 HF cows in Japan, using real-time PCR genotyping. The survey results are summarized in Table 1. Among the 48 Bali cattle, only 5 cows were heterozygous (A/G) for the A and G alleles; however, 36 cows and 7 males had only the G allele (G/G and G/-, respectively), resulting in a very high G allele frequency (0.944). Among the 5 Jaliteng cattle, 2 cows were heterozygous (A/G), resulting in a high G allele frequency (0.714). All 20 water buffaloes had only the G allele (G/G or G/-), resulting in a maximal G allele frequency (1) in this herd. In contrast, the A and G alleles were almost equally present in HF cows, resulting in a moderately high G allele frequency (0.514).

We evaluated the genotyping results of one to five individuals from each of the four assessed herds using Sanger sequencing. The genotyping results obtained by real-time PCR were consistent with the Sanger sequencing results from all herds used in this study (Figure 2). The representative real-time PCR amplification plots of the A/A, A/G, and G/G or G/- genotypes are shown in Appendix A.

### 3.2. AMH Concentration in Parous and Non-Parous HF Cows

We measured the serum AMH concentration in 108 HF cows, which comprised 69 parous and 39 non-parous cows. The AMH concentration was 477 ± 295 (mean ± standard deviation) and 506 ± 389 pg/mL in parous and non-parous cows, respectively. Based on the genotyping results, parous cows were further divided into 23 with genotype A/A, 27 with A/G, and 19 with G/G and non-parous cows into 7 with genotype A/A, 18 with A/G, and 14 with G/G. The AMH concentration was 629 ± 304, 424 ± 273, and 372 ± 254 pg/mL in parous cows with A/A, A/G, and G/G genotypes, respectively, and 585 ± 402, 422 ± 265, and 574 ± 507 pg/mL in non-parous cows with A/A, A/G, and G/G genotypes, respectively. The AMH concentrations are shown as box-and-whisker plots in parous (Figure 3A) and non-parous cows (Figure 3B).

The AMH levels differed significantly between the three genotypes of parous HF cows (*p* < 0.01; Friedman rank sum test). We thus further analyzed the differences in AMH concentrations in parous cows among the three genotypes using the Wilcoxon rank sum exact test with Bonferroni correction. The AMH concentrations in the A/G and G/G genotypes were significantly (*p* < 0.05) lower than those in the A/A genotype in parous cows (Figure 3A). In contrast, no significant differences were observed among the three genotypes of non-parous HF cows using either the Friedman rank sum test or the Wilcoxon rank sum exact test with Bonferroni correction (Figure 3B).

## 4. Discussion

Bali cattle (Figure 1A,B), a domestic bovine breed from Bali Island, Indonesia, is known for its adaptability and valuable contributions to regional agriculture [17]. Owing to their compact bodies, strong muscles, and upward-curving horns, they serve as draught animals and provide sought-after meat [18]. Thriving on limited resources and displaying disease resistance, these cattle play a vital role in Bali’s agriculture [19]. However, issues such as infertility and repeat breeding impede reproductive efficiency and productivity [20]. Implementing effective management strategies encompassing nutrition, healthcare, and genetic selection is considered essential to address these concerns and ensure the long-term sustainability of Bali cattle breeding programs.

Jaliteng cattle (Figure 1C,D), a subspecies of banteng native to Southeast Asia, are prized for their adaptability, genetic heritage, and cultural value [21,22,23]. Originating from crossbreeding Bali cattle with Javanese banteng bulls, these medium-to-large-sized animals possess distinctively curved horns [24]. They inherit favorable traits from their parent species, including meat quality and disease resistance, thereby promoting livestock productivity and genetic diversity. However, they are endangered because of habitat loss, hunting, and reproductive challenges [25,26]. Infertility and breeding issues among Jaliteng cattle have long been major concerns and require urgent attention for their conservation and management [27]. Preserving these unique animals is crucial for sustainable agriculture and the livelihoods of local farmers in Indonesia.

Water buffalo (Figure 1E) is a widespread bovine species in Asia [28]. Regarding their agricultural uses, water buffaloes are used for plowing fields, transporting goods, and supporting rural economies [29]. Water buffaloes provide milk with high fat and protein content, contributing to dairy production. They play important roles in meat production, food security, conservation, cultural symbolism, and environmental balance [30]. Despite their numerous benefits, buffalo breeding challenges persist because of factors such as low conception rates, infertility, and limited access to quality breeding bulls [31]. Addressing these issues requires improved reproductive techniques, veterinary care, and scientific investigations to optimize buffalo reproduction and increase herd sizes.

A previous GWAS identified a *FOXP3* gene variant (NC_037357.1: g.87298881A>G, rs135720414) associated with repeat breeding in JB cattle, indicating that the G allele is a variant associated with infertility [11]. In the present study, we examined herds of Bali and Jaliteng cattle, and water buffaloes, using real-time PCR genotyping targeting this variant. We demonstrated that the frequency of the risk-type G allele was markedly high in Bali (0.944) and Jaliteng (0.714) cattle and water buffaloes (1) and moderately high in HF cattle (0.514), as shown in Table 1. Our previous study reported that the G allele frequency ranged widely from 0.112 to 0.700 among several cattle breeds, including KH (0.112), JB (0.250), HF (0.466), and IM (0.700) cattle [12]. Among the breeds examined previously, IM, with the highest G allele frequency, is an Indonesian native bovine breed developed by crossing zebu (*B. indicus*) and banteng cattle, the herds of which are predominantly reared by small-scale farmers on Madura Island, East Java, Indonesia [32,33,34]. In our previous study, we speculated that the high G allele frequency in IM cattle may be attributed to the genetic contribution of banteng cattle [12]. In the present study, Bali and Jaliteng cattle native to banteng had a very high G frequency, which could also be attributed to the genetic contribution of banteng. Indonesian cattle and buffaloes are primarily bred through natural mating with local breeding bulls, which can result in inbreeding and the spread of single mutant genes within limited breeding areas [34]. These factors contribute to a reduction in genetic variation and an increase in the prevalence of specific undesirable alleles. Based on these data from banteng-related cattle breeds and water buffaloes, the G allele of the *FOXP3* gene may originally be a wild-type variant in cattle breeds and buffaloes, which is an undesirable trait for domestic animals. In contrast, modern taurine cattle, including beef (JB and KH) and dairy (HF) cattle, may have been unintentionally selected for the non-risk A allele during domestication and development [12], resulting in low-and-moderate G allele frequencies. Further studies are required to confirm these hypotheses by increasing the sample size of various bovine breeds, especially Asian old cattle breeds and water buffaloes.

Water buffaloes and cattle belong to the family Bovidae and subfamily Bovinae, with the genera *Bubalus* (buffalo) and *Bos* (cattle) belonging to the tribe Bovini [35]. The evolutionary history of buffaloes and cattle involves ancient hybridization events within their genomes [36]. Given that major and local Indonesian cattle breeds, including Bali, Jaliteng, and IM cattle, exhibit high frequencies of the G allele, the high G allele frequency observed in Indonesian water buffaloes may be influenced by their common ancestral origin and genetic hybridization, as observed between banteng and zebu cattle [37]. However, in the current study, we preliminarily examined a small herd of water buffaloes and obtained data based on a small population. Further research is required to confirm this hypothesis.

Among the taurine cattle breeds examined, HF dairy cattle had a moderate frequency of the G allele: 0.466 in our previous study and 0.514 in the current study, which was higher than those in beef cattle, including JB (0.250) and KH (0.112) [12]. The higher frequency of the G allele in HF dairy cattle may be because breeding programs for this breed have focused mainly on milk yield and quality rather than fertility. Previously, we investigated the relationship between the genotype of this variant and the reproductive performance in a small HF herd (19 cows) [38]. This study revealed that there are significant differences (*p* = 0.017) in the formation of follicular cysts (A/A: 0%, A/G: 61.5%, and G/G: 100%) among the three genotypes, and it confirmed a significant difference (*p* = 0.046) in the postpartum days open between cows with follicular cysts and those without cysts. These findings suggest that the G allele is associated with infertility in HF dairy cows; however, the limited number of cows in that study prevented this observation from being fully generalizable [38].

Therefore, in the present study, we measured serum AMH concentrations in a moderately sized HF herd (108 cows) and compared the results among the three genotypes after dividing them into 69 parous and 39 non-parous cows. The AMH concentration in parous cows with the A/A genotype was significantly higher than that in parous cows with the A/G and G/G genotypes (Figure 3A). Peripheral AMH concentrations are indicative of the size of the ovarian reserve and are promising biomarkers of fertility that could be utilized to improve breeding schemes for the reproductive performance of dairy cows [13]. Inflammation during the early postpartum period may decrease peripheral AMH levels, and subsequently affect the reproductive prognosis of postpartum HF cows [14]. Therefore, decreased AMH concentrations are associated with infertility in dairy cows. Based on our findings, HF cows carrying the G allele may be disadvantaged in terms of reproductive performance. However, in the present study, there were no significant differences among the three genotypes of non-parous HF cows (Figure 3B). This suggests that the combination of the G allele and aging may affect the fertility of dairy cows through multiple mechanisms, including decreased AMH concentrations as indicated in this study, increased number of follicular cysts [38], and postpartum inflammation [14]. Further studies are needed to resolve these issues by increasing the sample size from various bovine breeds and analyzing multivariable parameters such as age, parity number, and postpartum period that can influence AMH levels.

## 5. Conclusions

Our study found that the G allele frequency of the bovine *FOXP3* gene was markedly high in banteng-related cattle breeds, water buffaloes, and HF cattle. These data suggest that the G allele may be a wild-type variant in bovine species. We also found that the AMH concentration in the A/G and G/G genotypes was significantly lower than that in the A/A genotype in parous HF cows, suggesting that the G allele, combined with aging, is associated with infertility. Therefore, infertility in bovine herds may be improved by the selection and/or introduction of the A allele of the bovine *FOXP3* gene. In particular, the high frequency of the G allele in Bali, Jaliteng, and HF cattle suggests the need for focused efforts to address and reduce infertility issues in this population by using individuals carrying the A allele for breeding. However, the relationship among genetic background, aging, and reproductive disorders differs for each cattle breed and water buffalo; therefore, careful examinations are required for each individual breed. Further studies are necessary to advance breeding strategies and management practices for these valuable bovids.

## Figures and Tables

**Figure 1 animals-15-02407-f001:**
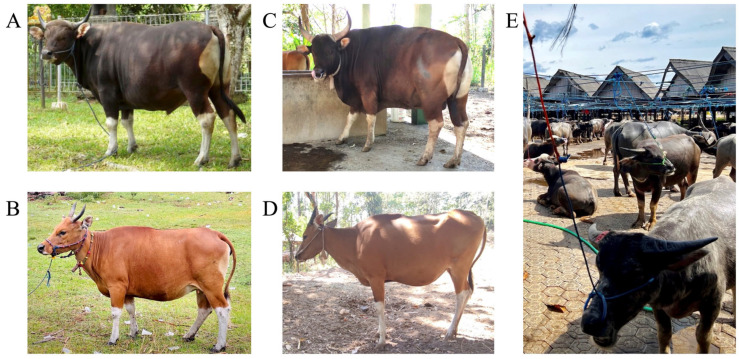
Representative appearances of Bali (*Bos javanicus domesticus*; (**A**), male; (**B**), female) and Jaliteng ((**C**), male; (**D**), female) cattle, a crossbreed between Bali and banteng cattle (*B. javanicus*), and water buffaloes (*Bubalus bubalis*, (**E**)), which were used in this study.

**Figure 2 animals-15-02407-f002:**
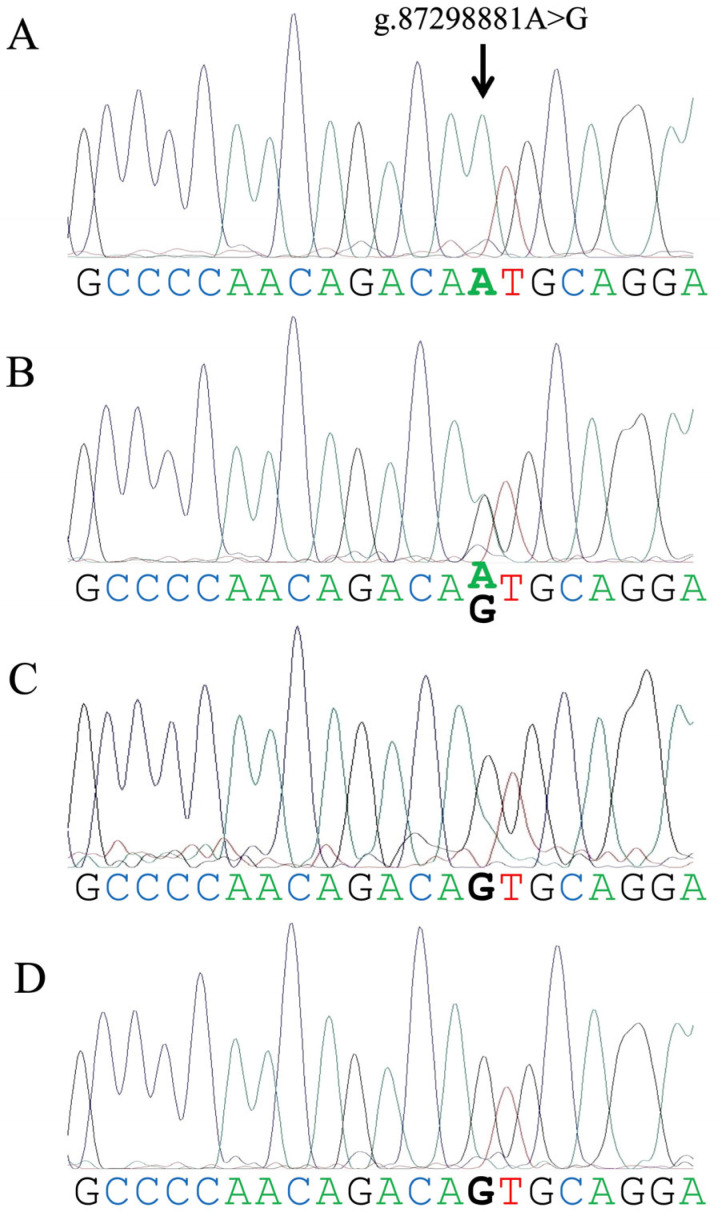
Representative Sanger sequencing electropherograms illustrating the A/A ((**A**), a Holstein Friesian cow), A/G ((**B**), a Bali cow), G/- ((**C**), a Jaliteng bull), and G/G ((**D**), a female water buffalo) genotypes associated with a single-nucleotide variant (arrow, g.87298881A>G) in the upstream of the bovine *FOXP3* gene.

**Figure 3 animals-15-02407-f003:**
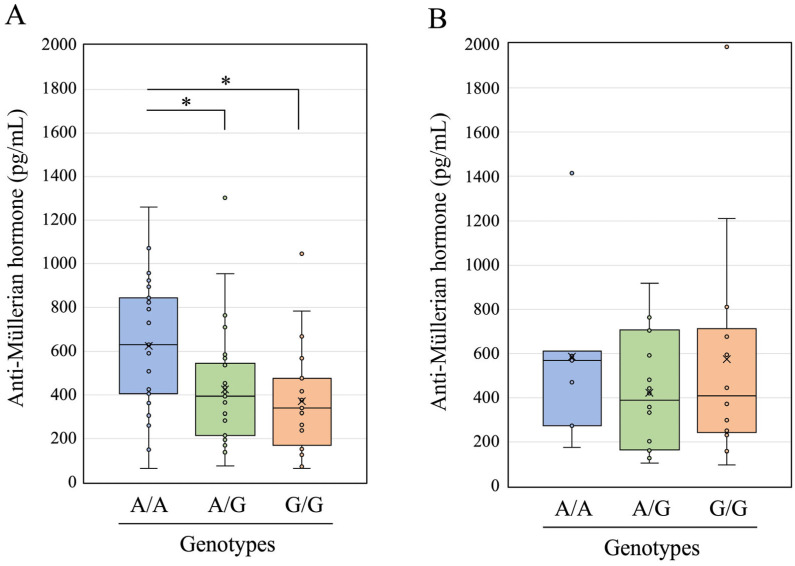
Anti-Müllerian hormone concentration in parous cows (**A**) and non-parous cows (**B**) with three different genotypes (A/A, A/G, and G/G) of a single-nucleotide variant (g.87298881A>G) upstream of the bovine *FOXP3* gene. * Differences in data between different groups were statistically significant (*p* < 0.05) using the Wilcoxon rank sum exact test with Bonferroni correction.

**Table 1 animals-15-02407-t001:** The numbers of cattle and water buffaloes genotyped and the frequencies of the bovine *FOXP3* gene variant.

Cattle and Buffalo	Examined Number		A/A or A/- *	A/G	G/G or G/- *	G Allele Frequency
Bali	Female	41	0	5	36	0.944
	Male	7	0	-	7	
Jaliteng	Female	2	0	2	0	0.714
	Male	3	0	-	3	
Water buffalo	Female	15	0	0	15	1
	Male	5	0	-	5	
Holstein Friesian	Female	108	30	45	33	0.514

* Genotypes of male hemizygote.

## Data Availability

The original contributions presented in this study are included in the article and Appendix A. Further inquiries can be directed to the corresponding author.

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
