# Peer review of "The High Frequency of a G-Allele Variant of the FOXP3 Gene in Old Asian Cattle Breeds, Water Buffaloes, and Holstein Friesian Cows: A Potential Link to Infertility"

_animals, 2025, doi:10.3390/ani15162407_

Round 1

Reviewer 1 Report

Comments and Suggestions for Authors

This manuscript authored by Faruq et al. presents an important investigation into the frequency of a FOXP3 gene variant (g.87298881A>G; rs135720414) across various bovine species, including Bali and Jaliteng cattle, water buffaloes, and Holstein Friesian cows, and its association with infertility. The study is timely, well-structured, and expands upon prior work in Japanese Black cattle by incorporating indigenous and under-studied breeds in Southeast Asia.

The novel insight that the G-allele is highly prevalent in banteng-related cattle and water buffalo populations, potentially representing a wild-type variant with detrimental effects on modern reproduction traits, adds valuable data to the field of bovine reproductive genetics.

However, the manuscript should address several limitations regarding sample size and the speculative interpretation of evolutionary origin. With minor revisions, this work will be a valuable contribution to the literature.

Specific Comments

  1. The Jaliteng cattle group included only five individuals (2 females). Conclusions about allele frequency in this population from such a small sample may be overstated. The reviewer recommends that this be acknowledged as a pilot observation and described as a limitation of the study in the Discussion section.
  2. While the analysis of AMH concentration across genotypes is statistically appropriate, the manuscript does not account for confounding variables such as age, parity number, and postpartum period, which can influence AMH levels. The reviewer recommends including a brief discussion on the lack of multivariate adjustment.

Author Response

Responses to the comments from Reviewer 1

Dear Reviewer 1,

First of all, we appreciate the time the editor and reviewers have taken to read and review our manuscript. Their valuable comments have significantly improved several aspects of our paper. The following document presents our responses to comments and suggestions from the reviewer. It includes the original comments in italics and blue and the subsequent responses we made. The revised parts are indicated in red.

Comments and Suggestions for Authors

This manuscript authored by Faruq et al. presents an important investigation into the frequency of a FOXP3 gene variant (g.87298881A>G; rs135720414) across various bovine species, including Bali and Jaliteng cattle, water buffaloes, and Holstein Friesian cows, and its association with infertility. The study is timely, well-structured, and expands upon prior work in Japanese Black cattle by incorporating indigenous and under-studied breeds in Southeast Asia.

Authors’ response: We thank the reviewer for understanding the objectives and strengths of our study. We appreciate it.

The novel insight that the G-allele is highly prevalent in banteng-related cattle and water buffalo populations, potentially representing a wild-type variant with detrimental effects on modern reproduction traits, adds valuable data to the field of bovine reproductive genetics.

Authors’ response: We thank the reviewer for highly evaluating our data in this study. We appreciate it.

However, the manuscript should address several limitations regarding sample size and the speculative interpretation of evolutionary origin. With minor revisions, this work will be a valuable contribution to the literature.

Authors’ response: We thank the reviewer for providing the valuable comments about the limitations of our study. We agree with the reviewer’s opinion. We will carefully revise our paper according to the reviewer's comments and suggestions as follows.

Specific Comments

The Jaliteng cattle group included only five individuals (2 females). Conclusions about allele frequency in this population from such a small sample may be overstated. The reviewer recommends that this be acknowledged as a pilot observation and described as a limitation of the study in the Discussion section.

Authors’ response: We thank the reviewer for the valuable comment. We agree with the reviewer’s opinion and added the description about the limitation of our study as the last sentence of the 4th paragraph in the Discussion section as follows.

"Further studies are required to confirm these hypotheses by increasing the sample size of various bovine breeds, especially Asian old cattle breeds and water buffaloes."

While the analysis of AMH concentration across genotypes is statistically appropriate, the manuscript does not account for confounding variables such as age, parity number, and postpartum period, which can influence AMH levels. The reviewer recommends including a brief discussion on the lack of multivariate adjustment.

Authors’ response: We thank the reviewer for the valuable comments. We also agree with the reviewer's opinion and added a phrase to the last sentence of the Discussion section as follows.

"Further studies are needed to resolve these issues by increasing the sample size from various bovine breeds and analyzing multivariable parameters such as age, parity number, and postpartum period that can influence AMH levels."

Reviewer 2 Report

Comments and Suggestions for Authors

Reproductive failure in cattle production is a global concern, and it is especially more common in buffaloes. Exploring the genetic causes related to reproductive failure is an important step toward identifying genetic parameters for selecting breeding animals at a very early age. In this regard, the authors have rightly chosen to study allele variants of the FOXP3 gene, which are associated with reproductive performance across different livestock species. Here, I have highlighted some issues to help improve the manuscript.
2. Materials and Methods 
2.1. Animals and blood collection
Line 117 - 121: The sentence "The body condition score (BCS) was... (mean 3.3) times in water buffalo." may be deleted, as BCS and parity are neither explained nor correlated with any findings in the results section.
2.2. DNA extraction and genotyping 
The authors used PCR, a genotyping assay, and PCR-based sequencing methods for genotyping the FOXP3 gene. However, each methodology has not been adequately explained. Information regarding the primers used, PCR conditions, and the genotyping assay procedure should be clearly described. Furthermore, simple PCR alone does not provide genotyping results; therefore, the authors should specify the exact procedures followed for genotyping.
Line 136: The phrase "Several DNA samples..." is vague. The authors should provide the exact number of samples used for genotyping.
 2.3. Measurement of AMH
Expand AMH fully
3. Results
3.1. Allele frequencies of the FOXP3 gene variant
Lines 159 and 168: The authors have presented genotyping results obtained using real-time PCR; however, the real-time PCR procedure has not been described in the methodology section. This should be corrected by clearly specifying the actual procedure used for genotyping. 
Furthermore, the authors are advised to provide supporting evidence for the genotyped samples, such as images of agarose gel electrophoresis (if applicable) or genotyping assay outputs, particularly allelic discrimination plots, to validate the authenticity of their results.
Table 1:
It is recommended that this table also include the genotypic frequencies for better clarity and to enhance the interpretability of the genotyping results.
3.2. AMH concentration in parous and non-parous HF cows
The authors have estimated AMH concentrations in parous and non-parous Holstein Friesian (HF) cows but not in other breeds. Given that the G allele frequency was found to be high in Bali cattle, it is unclear why AMH concentrations were not measured in this breed. It is therefore recommended that AMH levels be assessed in the remaining cattle breeds as well, particularly Bali cattle, to allow for a more comprehensive analysis. A comparative evaluation between Bali and HF cattle would provide more meaningful insights and should be included, supported with proper justification.

Additionally, the exact measured values from the AMH ELISA results should be reported in the manuscript for better clarity and reproducibility.

Regarding Figure 3, the AMH values are shown to differ significantly in parous HF cattle but not in non-parous HF cattle. While the results are explained separately, a logical and valuable extension would be to directly compare AMH concentrations between parous and non-parous HF cows. This would help determine the potential impact of the G allele on reproductive status across different physiological conditions within the same breed.
4. Discussion
Lines 200–227 (First Three Paragraphs):
The first three paragraphs in this section appear to contain introductory information related to the sample population used in the study. As such, they are more appropriate for inclusion in the Introduction section. Alternatively, this content may be briefly summarized to maintain focus and avoid redundancy within the Results and Discussion section.

5. Conclusions
Line 293:
The phrase "risk G allele frequency" is inappropriate, as the presence of the G allele has not been established as a risk factor. The word "risk" should be avoided in this context, and the sentence should be rephrased accordingly to reflect the actual findings.

Author Response

Responses to the comments from Reviewer 2

Dear Reviewer 2,

First of all, we appreciate the time the editor and reviewers have taken to read and review our manuscript. Their valuable comments have significantly improved several aspects of our paper. The following document presents our responses to comments and suggestions from the reviewer. It includes the original comments in italics and blue and the subsequent responses we made. The revised parts are indicated in red.

Comments and Suggestions for Authors

Reproductive failure in cattle production is a global concern, and it is especially more common in buffaloes. Exploring the genetic causes related to reproductive failure is an important step toward identifying genetic parameters for selecting breeding animals at a very early age. In this regard, the authors have rightly chosen to study allele variants of the FOXP3 gene, which are associated with reproductive performance across different livestock species. Here, I have highlighted some issues to help improve the manuscript.

Authors’ response: We thank the reviewer for the valuable comments. We revised our paper according to the reviewer's comments and suggestions as follows and explained the reasons in detail if we were not able to revise in some issues.

2. Materials and Methods

2.1. Animals and blood collection

Line 117 - 121: The sentence "The body condition score (BCS) was... (mean 3.3) times in water buffalo." may be deleted, as BCS and parity are neither explained nor correlated with any findings in the results section.

Authors’ response: We thank the reviewer for the advice. We deleted these two sentences and two references (nos. 15 and 16).

"The body condition score (BCS) was 2–5 (mean 3.5) in Bali cattle, 2.5–5 (mean 3.5) in Jaliteng cattle, and 2–5 (mean 3.9) in water buffaloes. The BCS was determined according to the criteria for Bali and Jaliteng cattle [15] and water buffaloes [16]." were deleted from the Materials and Methods section.

We are very sorry that the information about Holstein Friesian cows were missing in the original manuscript because of our careless mistake. We added a paragraph about the information of HF cows used in this study into 2.1. Animals and blood collection as follows.

"In 2019, blood samples were collected via jugular or caudal venipuncture from 69 parous and 39 non-parous HF cows housed in Fukuoka Prefecture, Japan. Blood was collected to measure AMH concentration and to genotype the FOXP3variant in parous cows 4 weeks after delivery and in non-parous cows during pregnancy. The parity of the parous cows ranged from 1 to 7 (average 3.2) times. For the measurement of AMH concentration, serum was obtained by centrifugation immediately after blood collection and was stored at -30 °C until measurement."

2.2. DNA extraction and genotyping

The authors used PCR, a genotyping assay, and PCR-based sequencing methods for genotyping the FOXP3 gene. However, each methodology has not been adequately explained. Information regarding the primers used, PCR conditions, and the genotyping assay procedure should be clearly described. Furthermore, simple PCR alone does not provide genotyping results; therefore, the authors should specify the exact procedures followed for genotyping.

Authors’ response: We thank the reviewer for pointing this out. According to the reviewer's request, we added the detailed description about the genotyping in 2.2. DNA extraction and genotyping as follows.

"Genotyping of the bovine FOXP3 gene variant (NC_037357.1: g.87298881A>G, rs135720414) was performed as previously described [12] using the extracted DNA as a template. PCR amplifications were performed using a StepOne Real-Time PCR System (Applied Biosystems, Foster City, CA, USA). Specific primer pairs (forward, CCATGTGGCTTCTGAGAAATAGTCA and reverse, TACCTGGAGGGCCAGACT) and TaqMan minor groove binder probes (A-allele, TCTTCCTGCATTGTCTG and G-allele, TCTTCCTGCACTGTCTG), linked to each fluorescent reporter dye (6-carboxyrhodamine and 6-carboxyfluorescein, respectively) at the 5'-end and a non-fluorescent quencher dye at the 3'-end, were used. Real-time PCR amplifications were carried out in a final volume of 5 µL consisting of 2× PCR master mix (TaqMan GTXpress Master Mix; Applied Biosystems) and 80× genotyping assay mix (TaqMan SNP Genotyping Assays; Applied Biosystems) containing the specific primers at 450 nM, TaqMan probes at 100 nM, and template DNA. A negative control containing nuclease-free water instead of the template DNA was included in each run. The cycling conditions were 20 s at 95 °C, followed by 50 cycles of 3 s at 95 °C and 20 s at 60 °C, with a subsequent holding stage at 25 °C for 30 s. The data obtained were analyzed using StepOne version 2.3 (Applied Biosystems)."

Line 136: The phrase "Several DNA samples..." is vague. The authors should provide the exact number of samples used for genotyping.

Authors’ response: We thank the reviewer for point this out. We added the exact number of samples used for Sanger sequencing and genotyping as follows.

"DNA samples with three genotypes (A/A, A/G, and G/G or G/-) from 1 HF (1 female), 5 Bali (2 males, 3 females), and 3 Jaliteng cattle (1 male, 2 females) and 2 water buffaloes (1 male, 1 female) were used to validate the genotyping assay, following genotype confirmation based on Sanger sequencing (Kazusa Genome Technologies Ltd., Kisarazu, Japan). For Sanger sequencing, PCR was performed using a forward primer (AGGGCTCAGATGCAGAC) and a reverse primer (GGATATGGTCTGTCTGGT), which produced a 166-base pair amplicon [12]. Genotyping using the real-time PCR assay was performed in 48 Bali cattle (7 males, 41 females), 5 Jaliteng cattle (3 males, 2 females), 20 water buffaloes (5 males, 15 females), and 108 HF cows."

2.3. Measurement of AMH

Expand AMH fully

Authors’ response: We thank the reviewer for point this out. We measured AMH concentration using a commercial ELISA kit according to the manufacturer's instructions. Therefore, we changed a phase "as described previously" to "according to the manufacturer's instructions" to avoid misunderstanding. Readers can measure AMH based on this commercial kit's instructions.

3. Results

3.1. Allele frequencies of the FOXP3 gene variant

Lines 159 and 168: The authors have presented genotyping results obtained using real-time PCR; however, the real-time PCR procedure has not been described in the methodology section. This should be corrected by clearly specifying the actual procedure used for genotyping.

Furthermore, the authors are advised to provide supporting evidence for the genotyped samples, such as images of agarose gel electrophoresis (if applicable) or genotyping assay outputs, particularly allelic discrimination plots, to validate the authenticity of their results.

Authors’ response: We appreciate the reviewer's valuable comments. We already added the detailed description about the genotyping method in the Materials and Methods section as mentioned above. In addition, we newly added Figure S1 that presents "Representative real-time PCR amplification plots of the A/A, A/G, and G/G or G/- genotypes ..." according to the reviewer's request and also added a sentence as the last sentence of the 2nd paragraph in the Results section as follows. Please see Figure S1. We agree that this figure can make our genotyping method more reliable. We appreciate it.

"The representative real-time PCR amplification plots of the A/A, A/G, and G/G or G/- genotypes were shown in Figure S1."

Table 1:

It is recommended that this table also include the genotypic frequencies for better clarity and to enhance the interpretability of the genotyping results.

Authors’ response: We thank the reviewer for this important comment. In Table 1, we showed only G-allele frequencies. The G-allele frequencies in Bali and Jaliteng cattle and water buffaloes are very high (close to 1), so readers can easily understand that A-allele frequencies are very low (close to 0). On the other hand, the data obtained from these three breeds have a limitation for interpretation because of the small or relatively small number of individuals. Therefore, I thought it would be better not to make any additional data and/or calculations that could bias the results. In contrast, the G-allele frequency of HF cows is about 0.5 suggesting A- and G-frequencies are almost same. Readers can easily imagine percentages of all genotypes because the number of individuals examined were close to 100. We thought that in this case of HF, genotypic frequencies and percentages were not necessary in Table 1. Additional data and calculations may be able to make the table look more complicated and moderately difficult to understand. We appreciate your understanding.

3.2. AMH concentration in parous and non-parous HF cows

The authors have estimated AMH concentrations in parous and non-parous Holstein Friesian (HF) cows but not in other breeds. Given that the G allele frequency was found to be high in Bali cattle, it is unclear why AMH concentrations were not measured in this breed. It is therefore recommended that AMH levels be assessed in the remaining cattle breeds as well, particularly Bali cattle, to allow for a more comprehensive analysis. A comparative evaluation between Bali and HF cattle would provide more meaningful insights and should be included, supported with proper justification.

Authors’ response: We really thank the reviewer for the valuable comments and recommendations. We agree with the reviewer's opinions. However, we collected blood samples from old Asian cattle and water buffaloes in local areas in Indonesia where no facilities for AMH measurement were available, but blood from HF cows in an urban area in Japan and we can easily access the facilities for AMH measurement. That is because we were not able to measure AMH concentrations in Asian old cattle breeds and water buffaloes. This was also a limitation of our present study.

AHM concentrations and their detailed reproductive performance data would be more interesting to further evaluate the infertility in Indonesian cattle. Therefore, we are now evaluating the simple test ion-chromatography kit for the measurement of AMH levels, which can be used even in local areas in Indonesia. So, we have a plan and are now preparing to evaluate the relationship between AMH and FOXP3 genotypes in Indonesian cattle breeds. We appreciate your understanding.

Additionally, the exact measured values from the AMH ELISA results should be reported in the manuscript for better clarity and reproducibility.

Authors’ response: We thank the reviewer for the important comment. We added the mean ± SD of AMH concentrations in total parous and non-parous cows and their groups with each genotype in the Results section as follows.

"We measured the serum AMH concentration in 108 HF cows, which comprised 69 parous and 39 non-parous cows. The AMH concentration was 477 ± 295 (mean ± standard deviation) and 506 ± 389 pg/mL in parous and non-parous cows, respectively. Based on the genotyping results, parous cows were further divided into 23 with genotype A/A, 27 with A/G, and 19 with G/G and non-parous cows into 7 with genotype A/A, 18 with A/G, and 14 with G/G. The AMH concentration was 629 ± 304, 424 ± 273, and 372 ± 254 pg/mL in parous cows with A/A, A/G, and G/G genotypes, respectively, and 585 ± 402, 422 ± 265, and 574 ± 507 pg/mL in non-parous cows with A/A, A/G, and G/G genotypes, respectively. The AMH concentrations are shown as box-and-whisker plots in parous (Figure 3A) and non-parous cows (Figure 3B)."

Regarding Figure 3, the AMH values are shown to differ significantly in parous HF cattle but not in non-parous HF cattle. While the results are explained separately, a logical and valuable extension would be to directly compare AMH concentrations between parous and non-parous HF cows. This would help determine the potential impact of the G allele on reproductive status across different physiological conditions within the same breed.

Authors’ response: We thank the reviewer for the important comment related to the above issue. Indeed, we statistically compared the difference of AMH concentration between parous and non-parous cows using whole data and data divided into genotypes, but there was no significant difference between them. Therefore, we did not mention such data. In the revised manuscript, the additional data of mean ± SD of AMH concentration can apparently show the similarity and broad deviation among all groups except for the difference between A/A and A/G or G/G in parous cows. Consequently, the actual exact measured values newly described in the text of our paper made the AMH data more informative in the revised manuscript. We appreciate your understanding.

4. Discussion

Lines 200–227 (First Three Paragraphs):

The first three paragraphs in this section appear to contain introductory information related to the sample population used in the study. As such, they are more appropriate for inclusion in the Introduction section. Alternatively, this content may be briefly summarized to maintain focus and avoid redundancy within the Results and Discussion section.

Authors’ response: We thank the reviewer for the important comment. We can understand what the reviewer suggests. However, we cannot agree partly with the reviewer's opinion in this issue. We explained that reason as follows.

When we started to write this paper, we thought of this issue very well by wondering which we should write this part in the Introduction or Discussion sections. Consequently, we decided to write this issue in the Discussion section because we wanted our research objectives clear and understandable in the Introduction section with a simple introduction of these cattle breeds. The relatively long description about rare Asian cattle breeds and water buffaloes in the Introduction section may make the Introduction busier and more difficult to understand the objective of our study smoothly.

However, we also thought that the relatively detailed explanations about Bali and Jaliteng cattle, and water buffaloes were necessary because only researchers related to Indonesia and east Asian countries know these cattle breeds. Our paper should provide the appropriate information of these rare Asian cattle breeds to all researchers and readers who do not know these cattle breeds. Therefore, we would like to keep these descriptions in the early part of the Discussion section. We appreciate your understanding.

5. Conclusions

Line 293:

The phrase "risk G allele frequency" is inappropriate, as the presence of the G allele has not been established as a risk factor. The word "risk" should be avoided in this context, and the sentence should be rephrased accordingly to reflect the actual findings.

Authors’ response: We thank the reviewer for the important comment and suggestion. We agree with the reviewer's opinion, and deleted the term "risk" used to directly explain the G allele's risk throughout the manuscript including this part.

Reviewer 3 Report

Comments and Suggestions for Authors

This article explores the high frequency of the G-allele variant of the FOXP3 gene in  old Asian cattle breeds, water buffaloes, and Holstein Friesian cows, along with its potential link to infertility. The article needs to better establish the biological basis for the association between FOXP3 variants and infertility. Methodological details such as sample size, genotyping methods, and infertility diagnosis criteria require clarification. The analytical approach should specify statistical methods and consider population structure. Interpretations should be cautious, avoiding overstatements, and account for species evolutionary differences.

  • How many cattle breeds are involved in the article, and is there a corresponding relationship between the listed charts and tables?
  • What is the logical relationship between FOXP3 and AMH? Additionally, in which region of the FOXP3 gene is the SNP locus located, and is it in the coding region?

Author Response

Responses to the comments from Reviewer 3

Dear Reviewer 3,

First of all, we appreciate the time the editor and reviewers have taken to read and review our manuscript. Their valuable comments have significantly improved several aspects of our paper. The following document presents our responses to comments and suggestions from the reviewer. It includes the original comments in italics and blue and the subsequent responses we made. The revised parts are indicated in red.

Comments and Suggestions for Authors

This article explores the high frequency of the G-allele variant of the FOXP3 gene in old Asian cattle breeds, water buffaloes, and Holstein Friesian cows, along with its potential link to infertility. The article needs to better establish the biological basis for the association between FOXP3 variants and infertility. Methodological details such as sample size, genotyping methods, and infertility diagnosis criteria require clarification. The analytical approach should specify statistical methods and consider population structure. Interpretations should be cautious, avoiding overstatements, and account for species evolutionary differences.

Authors’ response: We thank the reviewer for providing several valuable comments and suggestions.

Regarding the biological basis for the association and relationship between FOXP3 and infertility, we responded later to the reviewer's following question (the 3rd question) about AMH.

Regarding the methodological details, we are very sorry that the information about Holstein Friesian cows were missing in the original manuscript because of our careless mistake. We added a paragraph about the information of HF cows used in this study into 2.1. Animals and blood collection as follows.

"In 2019, blood samples were collected via jugular or caudal venipuncture from 69 parous and 39 non-parous HF cows housed in Fukuoka Prefecture, Japan. Blood was collected to measure AMH concentration and to genotype the FOXP3variant in parous cows 4 weeks after delivery and in non-parous cows during pregnancy. The parity of the parous cows ranged from 1 to 7 (average 3.2) times. For the measurement of AMH concentration, serum was obtained by centrifugation immediately after blood collection and was stored at -30 °C until measurement."

In addition, we clearly described the genotyping method and the number of cattle breeds used in this study according to the reviewer 1's request in 2.2. DNA extraction and genotyping as follows. We expect that this kind of revision will also make the reviewer 3 satisfied.

"Genotyping of the bovine FOXP3 gene variant (NC_037357.1: g.87298881A>G, rs135720414) was performed as previously described [12] using the extracted DNA as a template. PCR amplifications were performed using a StepOne Real-Time PCR System (Applied Biosystems, Foster City, CA, USA). Specific primer pairs (forward, CCATGTGGCTTCTGAGAAATAGTCA and reverse, TACCTGGAGGGCCAGACT) and TaqMan minor groove binder probes (A-allele, TCTTCCTGCATTGTCTG and G-allele, TCTTCCTGCACTGTCTG), linked to each fluorescent reporter dye (6-carboxyrhodamine and 6-carboxyfluorescein, respectively) at the 5'-end and a non-fluorescent quencher dye at the 3'-end, were used. Real-time PCR amplifications were carried out in a final volume of 5 µL consisting of 2× PCR master mix (TaqMan GTXpress Master Mix; Applied Biosystems) and 80× genotyping assay mix (TaqMan SNP Genotyping Assays; Applied Biosystems) containing the specific primers at 450 nM, TaqMan probes at 100 nM, and template DNA. A negative control containing nuclease-free water instead of the template DNA was included in each run. The cycling conditions were 20 s at 95 °C, followed by 50 cycles of 3 s at 95 °C and 20 s at 60 °C, with a subsequent holding stage at 25 °C for 30 s. The data obtained were analyzed using StepOne version 2.3 (Applied Biosystems)."

"DNA samples with three genotypes (A/A, A/G, and G/G or G/-) from 1 HF (1 female), 5 Bali (2 males, 3 females), and 3Jaliteng cattle (1 male, 2 females) and 2 water buffaloes (1 male, 1 female) were used to validate the genotyping assay, following genotype confirmation based on Sanger sequencing (Kazusa Genome Technologies Ltd., Kisarazu, Japan). For Sanger sequencing, PCR was performed using a forward primer (AGGGCTCAGATGCAGAC) and a reverse primer (GGATATGGTCTGTCTGGT), which produced a 166-base pair amplicon [12]. Genotyping using the real-time PCR assay was performed in 48 Bali cattle (7 males, 41 females), 5 Jaliteng cattle (3 males, 2 females), 20 water buffaloes (5 males, 15 females), and 108 HF cows."

Regarding the infertility diagnosis criteria, we did not determine the criteria exactly and just utilized AMH concentration in HF cattle. This is a limitation of our study and population structure such a small number of Jaliteng and water buffaloes is also a weak point in our study. Therefore, we lightened our overstatement by describing a limitation of our study. We described those in the Discussion section according to the reviewer 1's requests, too.

We added the description about a limitation of our study as the last sentence of the 4th paragraph in the Discussion section as follows.

"Further studies are required to confirm these hypotheses by increasing the sample size of various bovine breeds, especially Asian old cattle breeds and water buffaloes."

In addition, we added a phrase to the last sentence of the Discussion section as follows.

"Further studies are needed to resolve these issues by increasing the sample size from various bovine breeds and analyzing multivariable parameters such as age, parity number, and postpartum period that can influence AMH levels."

How many cattle breeds are involved in the article, and is there a corresponding relationship between the listed charts and tables?

Authors’ response: We thank the reviewer for this important comment. We responded to this comment and described the exact number of cattle breeds and water buffaloes examined in this study when we responded to the reviewer's comment above. Please see above.

What is the logical relationship between FOXP3 and AMH?

Authors’ response: We thank the reviewer for the valuable comment. We discussed AMH in the last paragraph of the Discussion section. We mentioned by citing a paper that "Inflammation during the early postpartum period may decrease peripheral AMH levels, and subsequently affecting the reproductive prognosis of postpartum HF cows [14]". Inflammation caused by certain bacterial infections is likely to happen based on a decreased level of immune system. We think that a decreased level of immune system is caused partly by G-allele of FOXP3 and aging. This may be one of the logical relationships between FOXP3 and AMH (infertility); however, we do not want to overstate the logical relationship based on our current data. We appreciate your understanding.

Additionally, in which region of the FOXP3 gene is the SNP locus located, and is it in the coding region?

Authors’ response: We thank the reviewer for an important question. As we mentioned in the Introduction section, this X-linked single nucleotide variant (NC_037357.1: g.87298881A>G, rs135720414) is located 2175 base pairs upstream of the start codon of the bovine FOXP3 gene. Therefore, this variant is not in the cording region of the FOXP3 gene. According to the original paper that identified this variant (ref. no. 11), reporter assay results revealed that the activity of the FOXP3 promoter was lower in reporter constructs with the G-allele than in those with the A- allele by approximately 0.68 fold, suggesting the G-allele decreases FOXP3 transcription, in turn, can reduce the number of maternal Treg cells and lead to infertility.

In this study, we investigated the allele frequencies in several cattle breeds and the relationship between AMH and the genotypes in Holstein cows, but did not examine the FOXP3 transcription and Treg number and activity. Therefore, we did not discuss those issues to prevent the overstatement. We appreciate your understanding.

Round 2

Reviewer 2 Report

Comments and Suggestions for Authors

The authors have given valuable inputs pertaining to my comments; the article may be accepted for publication.

Author Response

Responses to the comments from Reviewer 2

Comments and Suggestions for Authors

The authors have given valuable inputs pertaining to my comments; the article may be accepted for publication.

Authors’ response: We appreciate the time the reviewer has taken to read and review our manuscript again. We are glad that the reviewer found our revised paper accepted for publication. The valuable comments and suggestions previously indicated by the reviewer have significantly improved our paper.

Reviewer 3 Report

Comments and Suggestions for Authors The manuscript has shown certain improvements compared to the previous version, which is commendable. However, there are still two issues that need to be addressed.​ This paper focuses on exploring the potential relationship between the frequency of a G-allele variant of FOXP3 and infertility, and the study design involves two cattle breeds and their crossbred offspring. Additionally, the sampling spans a large number of years, which raises concerns about whether the determination of AMH might be affected by age factors.​ Therefore, the article needs to explain the following issues:​
  1. Whether there is a relationship between cattle breeds and the incidence of infertility.​
  1. Whether there is a relationship between cattle age and the incidence of infertility.

Author Response

Responses to the comments from Reviewer 3

Comments and Suggestions for Authors

The manuscript has shown certain improvements compared to the previous version, which is commendable. However, there are still two issues that need to be addressed. This paper focuses on exploring the potential relationship between the frequency of a G-allele variant of FOXP3 and infertility, and the study design involves two cattle breeds and their crossbred offspring. Additionally, the sampling spans a large number of years, which raises concerns about whether the determination of AMH might be affected by age factors. Therefore, the article needs to explain the following issues:

Whether there is a relationship between cattle breeds and the incidence of infertility.

Whether there is a relationship between cattle age and the incidence of infertility.

Authors’ response: We appreciate the time the reviewer has taken to read and review our manuscript again. We also thank the reviewer for the above valuable comments and suggestions. We totally agree with these opinions. As we mentioned before and the reviewers also pointed out, there is a limitation in our paper because of a small number of individuals in a part of cattle breeds and water buffaloes used in this study. We cannot explain clearly the relationship among genetic background, aging, and reproductive disorders of all the breeds in the same way by only G-allele of FOXP3 gene. According to the reviewer's requests, we decided to add a new sentence in the Conclusion section showing the limitation of these issues as follows. We appreciate the reviewer's understanding.

"In particular, the high frequency of the G allele in Bali, Jaliteng, and HF cattle suggests the need for focused efforts to address and reduce infertility issues in this population by using individuals carrying the A allele for breeding. However, the relationship among genetic background, aging, and reproductive disorders differs for each cattle breed and water buffalo; therefore, careful examinations are required for each individual breed. Further studies are necessary to advance breeding strategies and management practices for these valuable bovids."